# Associations of Dynapenic Obesity and Sarcopenic Obesity with the Risk of Complications in COVID-19

**DOI:** 10.3390/ijms23158277

**Published:** 2022-07-27

**Authors:** Laura Pérez-Campos Mayoral, Carlos Alberto Matias-Cervantes, Eduardo Pérez-Campos, Carlos Romero Díaz, Luis Ángel Laguna Barrios, María del Socorro Pina Canseco, Margarito Martínez Cruz, Eduardo Pérez-Campos Mayoral, Carlos Josué Solórzano Mata, Francisco Javier Rodal Canales, Héctor Martínez Ruíz, María Teresa Hernández-Huerta

**Affiliations:** 1Centro de Investigación Facultad de Medicina UNAM-UABJO, Facultad de Medicina y Cirugía, Universidad Autónoma “Benito Juárez” de Oaxaca, Oaxaca City 68020, Mexico; laupcm9@gmail.com (L.P.-C.M.); lalb2906@gmail.com (L.Á.L.B.); socopina12@hotmail.com (M.d.S.P.C.); epcm@live.com.mx (E.P.-C.M.); universidad99@hotmail.com (C.J.S.M.); rrodal22@gmail.com (F.J.R.C.); drheccctor@hotmail.com (H.M.R.); 2CONACyT, Facultad de Medicina y Cirugía, Universidad Autónoma “Benito Juárez” de Oaxaca, Oaxaca City 68020, Mexico; carloscervantes.ox@outlook.com (C.A.M.-C.); carlos.rom.74he@gmail.com (C.R.D.); 3Tecnológico Nacional de México/IT Oaxaca, Oaxaca City 68030, Mexico; martinezcu9@hotmail.com; 4Facultad de Odontología, Universidad Autónoma “Benito Juárez” de Oaxaca, Oaxaca City 68120, Mexico

**Keywords:** sarcopenic obesity, ageing, COVID-19, complications, dynapenic obesity

## Abstract

Ageing is associated with changes in body composition, such as low muscle mass (sarcopenia), decreased grip strength or physical function (dynapenia), and accumulation of fat mass. When the accumulation of fat mass synergistically accompanies low muscle mass or reduced grip strength, it results in sarcopenic obesity and dynapenic obesity, respectively. These types of obesity contribute to the increased risk of cardiovascular disease and mortality in the elderly, which could increase the damage caused by COVID-19. In this review, we associated factors that could generate a higher risk of COVID-19 complications in dynapenic obesity and sarcopenic obesity. For example, skeletal muscle regulates the expression of inflammatory cytokines and supports metabolic stress in pulmonary disease; hence, the presence of dynapenic obesity or sarcopenic obesity could be related to a poor prognosis in COVID-19 patients.

## 1. Introduction

Obesity is a very serious medical condition worldwide. It is associated with many health complications such as cardiovascular disease, diabetes, musculoskeletal disorders, and cancer [1]. The phenotype for sarcopenic obesity is associated with an increase in respiratory disease and mortality [2]; its prevalence varies according to the approach used to treat it and the studied population. It has been reported to occur in a range of 2% to 42.9% of the population [3,4,5,6]. Considering that in clinical practice, functional status is more important than low muscle mass, here we defined dynapenic obesity (DO) as low muscle strength and high fat mass [7], and for sarcopenic obesity (SO) in the elderly, we use the body mass index (BMI) ≥ 30 Kg/m^2^ with gender specificity in the tertile of grip strength.

For its part, coronavirus disease-2019 (COVID-19) is an infectious disease caused by SARS-CoV-2 coronavirus that first broke out in Wuhan, China [8]. It can manifest as asymptomatic or with severe symptoms, partly depending on age, physical activity, nutrition, and associated comorbidities [9]. It commonly starts as a respiratory infection similar to a cold but can have severe symptoms such as fever, dry cough, and difficulty breathing; it can take between 2–14 days to appear after exposure [10]. Old age, obesity, hypertension, diabetes, and ethnic group are the main risk factors for infection and hospitalization by COVID-19 [11]; in addition, cardiovascular diseases increase complications and the possibility of death [12]. Furthermore, the presence of SO is related to a poor prognosis in patients with COVID-19 [13]. The aim of this review was to propose the mechanisms of comorbidity risk of COVID-19 patients with sarcopenic obesity or dynapenic obesity.

## 2. Sarcopenic Obesity and Dynapenic Obesity

Obesity is a chronic disease [14] defined as the abnormal or excessive accumulation of fat [15]. It is expressed in various phenotypes, one of which is sarcopenic obesity [1,16]. The most widely used means of identifying obesity is to calculate the body mass index, taking the weight in kilograms, and dividing by the height in meters squared. Adult obesity is defined as a BMI of ≥30 Kg/m^2^ [5]. However, due to the endocrine and inflammatory role of adipose tissue, it is also necessary to classify obese conditions based on the distribution and composition of body fat. For this, some phenotypes of obesity have been described: normal weight obese, metabolically obese normal weight, metabolically healthy obese, metabolically unhealthy obese, and sarcopenic obese [1,17]. In older people, different phenotypes have been reported as: nonobese nondynapenic, overweight nondynapenic, obese nondynapenic, sarcopenic obese, overweight sarcopenic, nonobese dynapenic, and dynapenic obese [18,19,20] (Figure 1).

It is well known that sarcopenia and obesity disorders are dependent on nutritional status [21], body composition, hormonal changes [22], strength and muscle mass [23], all of which act synergistically to increase the risk of disability [13]. 

Sarcopenic obesity (SO) is defined as obesity with the loss of muscle mass (ASMM/h^2^: < 5.18 Kg/m^2^ in women and <7.00 Kg/m^2^ in men), while dynapenic obesity (DO) is defined by the association of loss of leg muscle strength (<12 Kg women and <21 Kg men) or physical function; both phenotypes with an accumulation of body fat mass (>40% women and >28% men) or BMI ≥ 30 [24], particularly in those with comorbid diseases such as diabetes, arthritis, and cardiovascular and respiratory conditions [25,26,27]. It is important to mention that low muscle mass is associated with dynapenia and decreased motor capacity [28]. For the diagnosis of obesity in geriatric patients, the concept of DO is rarely considered; thus, patients with decreased muscle mass or low handgrip strength in sarcopenia may be included [29,30]. 

There are several mechanisms and factors related to the pathogenesis of SO and DO (Figure 2), such as (1) adipose tissue dysfunction characterized by adipocyte hyperplasia and hypertrophy [31,32]; (2) perilipins 5 is related to a decrease in the lipotoxicity and insulin resistance [33]; (3) systemic chronic sterile low-grade inflammation [34,35,36]; (4) vitamin D deficiency associated with handgrip strength but not with muscle mass [37]; (5) vitamin D receptor gene polymorphism of Fok1 associated with sarcopenia, lower gait speed, and lower handgrip strength [38]; (6) adipose tissue inflammation with an accumulation of macrophages and lymphocytes [39,40]; (7) during inflammation of adipose tissue, the accumulation of M1 macrophages around necrotic adipocytes produces the release of fatty acids. This is associated with the production of a greater amount of tumor necrosis factor-alpha (TNF-α), which releases more fatty acids from adipocytes, becoming a vicious circle that maintains the proinflammatory environment [41].

All these factors cause an asymptomatic inflammatory condition in hypertrophied adipose tissue with a high number of inflammatory cells, production of adipokines and other inflammatory cytokines [42]. The production of inflammatory cytokines motivates the arrival of immune cells, mainly macrophages, interferon gamma-producing TH1 lymphocytes (INF-γ), and CD^8+^ lymphocytes capable of initiating the inflammatory response [43,44].

In the skeletal muscle, fat droplets are accumulated as intermuscular adipose tissue and intramyocellular lipids (IMCLs) [45]. One characteristic of IMCLs is that they can induce a lipotoxic effect on muscles, which is characterized by impaired single-fiber contractility, leading to lower muscle strength and power in the elderly. This occurs because of the autophagy of muscle cells [46]. In the pathogenesis of sarcopenia, an important molecule identified at the neuromuscular junction is the C-terminal agrin fragment, which causes an age-dependent increase and muscle dysfunction [47]. Furthermore, during ageing, the change in muscle mass and weight gain (by lean mass) reflect the decrease in metabolic rate [48]. In addition, SO is related to elevated levels of IL-6, high-sensitivity C-reactive protein (hs-CRP) [49], IL-1 receptor antagonist, and soluble IL-6 receptor; all of these could contribute to apoptosis in myocytes and lead to a decrease in muscle mass and strength [50,51]. Thus, in SO, frailty and changes in immune function with age (immunosenescence) [52] are associated with physical inactivity and the reduction of energy expenditure, as well as impairment of movement and respiratory problems linked to metabolic alterations, leading to increased risk of comorbidity. 

The renin–angiotensin system (RAS) participates in the regulation of the cardiovascular-renal system and hydroelectrolyte balance; it also influences the heart, kidney, brain, and other tissues [53]. RAS is composed of a series of reactions that result in the formation of angiotensin II (Ang II) by the angiotensin-converting enzyme 2 (ACE2), whose actions are mediated by metabotropic receptors associated with G proteins, type 1 (AT1) and type 2 (AT2) [54,55]; these participate on the regulation of various processes such as vasoconstriction, water and sodium retention, and cell proliferation [56]. Several investigations indicate that the activation of the classical RAS, represented by ACE2-Ang II-AT1, has an important role in the deterioration of skeletal muscle because RAS signaling promotes skeletal muscle atrophy [57] and fibrosis [58], as well as insulin sensitivity [59]. SARS-CoV-2 uses ACE2 to enter host cells; thus, high expression of ACE2 is related to the disease severity [53,60]. Therefore, the use of ACE2 inhibitors and angiotensin receptor blockers (ARBs) could improve muscle performance decline and reduce frailty in sarcopenic obesity [61], while preventing and treating COVID-19 [62].

Vitamin D deficiency has also been found to be common in obese people [63]. This deficiency is associated with an increased risk of frailty, falls, and increased fracture risk in DO and SO [7,64,65]. Nevertheless, there are discrepancies in different disorder studies in which vitamin D is supplemented [36]. It should be remembered that vitamin D aids the body to absorb calcium, one of the main nutrients necessary for strong bones; we think that vitamin D levels and vitamin D receptor gene polymorphism should be taken into consideration in SO and DO.

Although the affectations in SO and DO vary from one individual to another, there is a consensus on the higher risk of mortality in older adults in both cases [2]. Berens et al. evaluated the possible association between mortality and obesity in people, both with and without sarcopenia, and found that 75-year-old women with SO have a greater risk of dying at 10 years, compared to those without sarcopenia or obesity, while for 87-year-old obese men without sarcopenia, it was associated with a survival limit of up to four years. [66].

## 3. Dynapenic and Sarcopenic Obesity and Their Association with Cardiovascular Disease Risk

Cardiovascular diseases are the main cause of death in the world, according to reports of the World Health Organization, more than 17.9 million deaths are estimated per year [67]. Cardiovascular diseases (CVDs) include disorders related to problems of the heart, blood vessels, and cardiac degeneration, among others. CVDs are associated with increased oxidative and inflammatory stress, as well as increased cell death [68]. Ageing is represented as the greatest risk factor for cardiovascular diseases, with approximately 40% of deaths reported in the elderly [69]. 

During ageing, there is a decrease in the activity of the physiological processes that allow the correct maintenance of the organism, which mainly affects cardiovascular tissues and increases the possibility of suffering from CVDs [69]. Alterations in the heart and vasculature due to ageing compromise the maintenance and proper function of the heart and arterial system, leading to structural changes in the heart, such as hypertrophy, decreased heart rate, increased arrythmias, apoptotic/necrotic cells, fibrosis, ischemic tissue, infiltrating of smooth-muscle cells, atherosclerotic plaque, ischemic tissue, and increased arterial stiffness [70]. This induces molecular changes, increase of reactive oxygen species, p53, p21, p16, β-galactosidase activity, 8-oxoguanine, and phosphorylation of γ-H2Ax; in turn, that decreases endothelial nitric oxide synthetase and nitric oxide, among others [71,72], related to hypertension, atherosclerosis, stroke arterial fibrillation, ischemia, and metabolic disease [70,71]. 

Other changes in ageing are a decrease in physical activity and an inadequate diet, such as a high carbohydrate intake with low protein intake, favoring a state of sarcopenic obesity that predisposes people to the development of CVDs [73,74]. 

Increased cardiovascular risk has been observed in DO and SO, particularly in sarcopenic obesity, when compared to obesity or sarcopenia [75,76]. Serum concentrations of hs-CRP, low-density lipoprotein cholesterol (LDL-C), soluble intercellular adhesion molecule type 1 (sICAM-1), and triglyceride are higher in SO and DO patients than in nonsarcopenic nonobese or nondynapenic nonobese patients [77]. hs-CRP is a systemic inflammation marker that predicts future cardiac events [78]. The risk of myocardial infarction is increased in lean sarcopenic patients and sarcopenic overweight, or obese patients compared to lean nonsarcopenic patients [19]. Furthermore, SO patients were associated with a fivefold-increased risk of developing atrial fibrillation [19]. 

SO is also associated with coronary artery calcification, independent of known risk factors for coronary artery disease, such as age, sex, hypertension, diabetes, dyslipidemia, and creatinine [79]. The combination of obesity and sarcopenia may be associated with an increased risk of coronary atherosclerosis, which can eventually lead to cardiovascular events. 

Pathogenetic mechanisms related to SO and CVDs could lead to a reduction in muscle mass and an accumulation of fat in muscle tissue that promotes a proinflammatory cascade and oxidative stress, as consequence stimulating mitochondrial dysfunction, muscle atrophy and insulin resistance [29]. Insulin resistance differentially affects the phosphatidylinositol 3-kinase and mitogen-activated protein kinase signaling pathways [80] and activates inflammatory pathways including IκB/nuclear factor κB (NF-κB), and c-Jun N-terminal kinase, which contributes to the development of atherosclerotic cardiovascular disease [81]. The simultaneous presence of sarcopenia and obesity is associated with both oxidative stress and a proinflammatory state (as indicated by high levels of TNF-α, IL-1b, IL-6, IL-8, IL-12, and hs-CRP), is also associated with an increased risk of CVD [82,83]; this excess of cytokines decreases muscle anabolism by facilitating muscle atrophy, modifying the function and proliferation of immune cells [84], which may explain the elevated levels of IL-6 and Th17 cells in SO patients with COVID-19. 

Although there are fewer studies specific of dynapenic obesity, dynapenic abdominal obese individuals have a higher prevalence of metabolic syndrome and lipid disorders, such as low levels of HDL-cholesterol, hypertriglyceridemia, hyperglycemia and high levels of glycosylated hemoglobin [85]. 

Likewise, a high BMI is associated with higher CVD mortality, although it is known that BMI is inversely correlated with the mortality rate in patients with coronary artery disease, the so-called “obesity paradox” has been also observed. In older adults or patients with several chronic diseases it is associated with an apparent decrease in cardiovascular adverse events; however, the “obesity paradox” could be a misclassification bias caused by use of BMI, or a way of selection named collider stratification bias [86,87]. Therefore, to assess visceral obesity, waist circumference (WC), the ratio of waist-to-hip circumferences or waist-to-hip ratio (WHR), or visceral adiposity index [88] measures are recommended since these correlate better with cardiovascular risk than BMI; that is because BMI represents fat mass and lean mass [89]. WC is related to an excess of abdominal fat, even in subjects with a normal BMI, it is associated with cardiometabolic diseases and is predictive of mortality [90].

According to studies of obesity phenotypes, metabolically healthy normal weight (MHNW) people who have sarcopenia are at high risk of cardiovascular disease. The presence of sarcopenia in other subtypes of obesity also increases the risk of CVD [91]. Sarcopenia when assessed by total skeletal muscle (total SM) cross-sectional area and index (divided by height squared) of the chest, a computed tomography scan, age, diabetes mellitus, and hypertension are independent predictors of mortality, in-hospital in COVID-19 patients [92]. Similarly, the metabolic consequences of obesity increase the risk of ischemic stroke [93], which in COVID-19 patients is associated with more severe infectious disease [94]. 

A stroke is defined as an ischemic or hemorrhagic cerebral infarction. It has been seen that in patients who have suffered from a stroke, a muscle-mass and muscle-strength reduction can occur, in addition to a fat deposition, which induces sarcopenia at a rate of 14 to 54% [95]. Although sarcopenia is commonly observed to be induced by stroke, pre-stroke sarcopenia is an independent predictor of stroke severity and is associated with poor functional outcomes and risks of malnutrition [96]. 

On the other hand, there is a paradox in incident stroke patients associated with obesity or overweight. In these patients, the prognosis is more favorable for major adverse cardiovascular events, such as coronary heart disease, recurrent stroke, peripheral vascular disease, heart failure, and cardiovascular-related mortality [97]. In the treatment of these patients with obesity and CVDs, semaglutide has been evaluated by the Semaglutide Effects on Cardiovascular Outcomes in People with Overweight or Obesity (SELECT) study [98]. Similarly, semaglutide and other antidiabetic drugs, as well as omega-3 fatty acids, are being evaluated for sarcopenia [99,100].

## 4. Sarcopenic Obesity, Dynapenic Obesity, and COVID-19

SO could increase the risk of severe complications and adverse outcomes in COVID-19 (Figure 3) [13,101]. Among the most notable findings of the COVID-19 patient are the effects of myalgia arthralgia, back pain, fatigue, and loss of grip strength [102], in addition to the fact that the SARS-CoV-2 infection mainly affects the epithelium of the lungs. Furthermore, other systems have also been involved, such as the immune [103], integumentary [104], neurological [105], digestive [106], genitourinary [107], cardiovascular [108], hematological [109], reproductive, and hormonal systems [110,111], causing very varied symptomatology. In the most severe cases, COVID-19 causes pneumonia, heart and kidney failure, and liver injury, thrombosis, shock, and even death. Twenty-five per cent of these patients develop severe lung disease that progresses to adult respiratory distress syndrome [112,113]. 

Adipose tissue could function as a deposit for a wider viral spread with increased immune activation and cytokine amplification in patients associated with abnormal cytokine profiles [114]. SARS-CoV-2 infection depends on its binding to target cells facilitated by ACE2, which is expressed in various human tissue [115], particularly in the lungs, bowels, kidneys, and blood vessels [116].

SO patients manifest a higher prevalence of type 2 diabetes and hypertension than those without sarcopenic [117,118], associating them with a higher risk of respiratory disease and mortality [2]. In addition to this, a Chinese study found that people aged 65 years or older with type 2 diabetes are more susceptible to COVID-19 [119,120]. 

In COVID-19 patients, an association has been observed between decreased skeletal muscle area, low skeletal muscle radiodensity, and increased complications during their stay in the intensive care unit [121]. Likewise, low muscle quality and ectopic fat accumulation lead to invasive mechanical ventilation complications or even death, while increased muscle density is a protective factor [122].

COVID-19 affects immune cells and the expression of inflammatory molecules that increases with disease severity [123]. Lung infections by SARS-CoV-2 could lead to elevated blood sugar levels by adipose dysfunction in adiponectin and adiponectin/leptin ratios [124,125], making it difficult to control infections and metabolic diseases with sarcopenia or obesity [126]. Wilkinson et al. found that SO patients are approximately 2.6 times more likely to have severe COVID-19 infection than obese patients; however, sarcopenia alone did not increase the risk of severe COVID-19 [127]. 

Immunosenescence affects both the innate and adaptive immune response [128], leads to increased susceptibility to infections, reduces vaccination responses in frail elderly people, and increases the risk of chronic inflammatory diseases [52,129]. In COVID-19, not only do factors such as smoking, hypertension, diabetes mellitus, chronic obstructive pulmonary disease, physical frailty, and C-reactive protein impact the severity/mortality of COVID-19, but also the components of DO, such as loss of grip strength and sarcopenia [130].

## 5. Conclusions 

The presence of dynapenia or sarcopenia aggravates multisystemic disease in SARS-CoV-2 infection; likewise, the combination of obesity with loss of grip strength or muscle mass due to ageing increases the risk of complications during hospitalization for COVID-19; which can affect mortality in the elderly as a result of alterations mainly in glucose metabolism and the respiratory, cardiovascular, and immune systems.

## Figures and Tables

**Figure 1 ijms-23-08277-f001:**
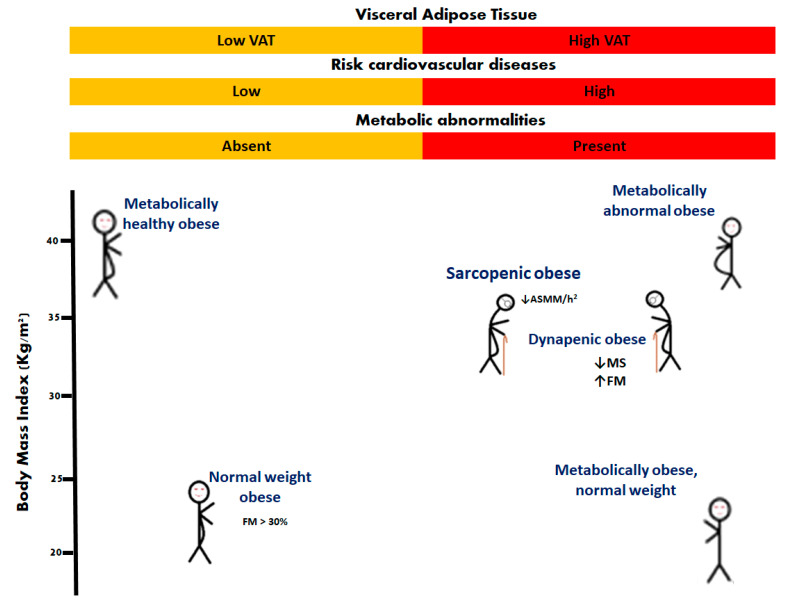
Phenotypes of obesity. In older people, sarcopenic obesity and dynapenic obesity are related to worsening disability. Appendicular skeletal muscle mass divided by height squared (ASMM/h^2^); fat mass (FM); muscle strength (MS).

**Figure 2 ijms-23-08277-f002:**
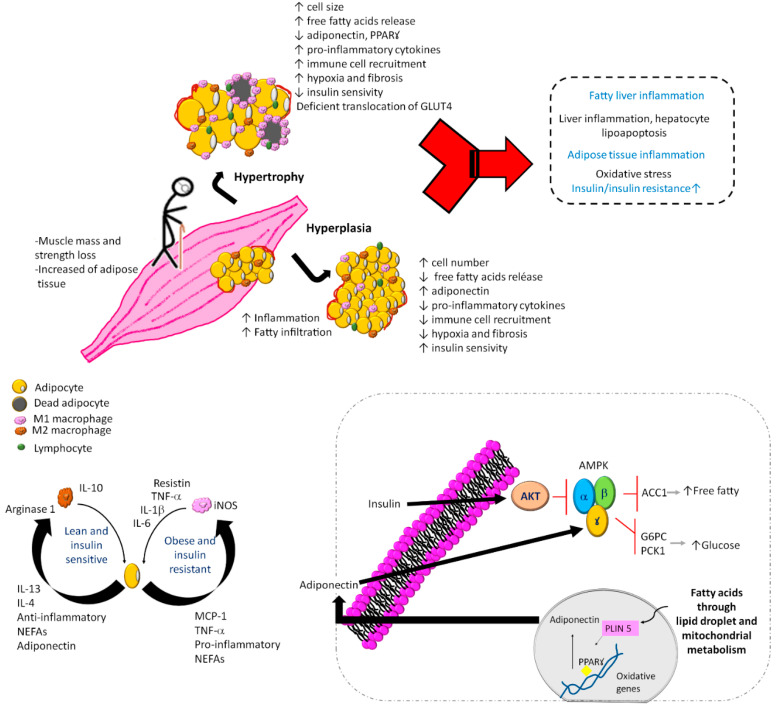
Mechanisms and factors related to the pathogenesis of sarcopenic obesity and dynapenic obesity. In sarcopenic obesity and dynapenic obesity, the muscle mass and strength loss with increased adipose tissue induces an inflammatory cascade and accumulation of immune cells, as well as leukocyte activation, adipogenesis, and adipocyte death. Added to physical inactivity, carbohydrate overload and lower protein intake cause a vicious circle of insulin resistance, where there is an increase in free fatty acids and M1 macrophages with alterations in mitochondrial metabolism by inactivation of regulators of energy homeostasis and inducers of regulated fatty acid oxidation, vitamin D deficiency, and D receptor gene polymorphism. G6PC, glucose-6-phosphatase; PKC1, phosphoenolpyruvate carboxykinase 1; AMPK, AMP-activated protein kinase; ACC1, acetyl-CoA carboxylase 1; NEFAs, nonesterified fatty acids; PLIN 5, perilipin 5; PPARꙋ, peroxisome proliferator-activated receptor gamma; GLUT4, glucose transporter type 4.

**Figure 3 ijms-23-08277-f003:**
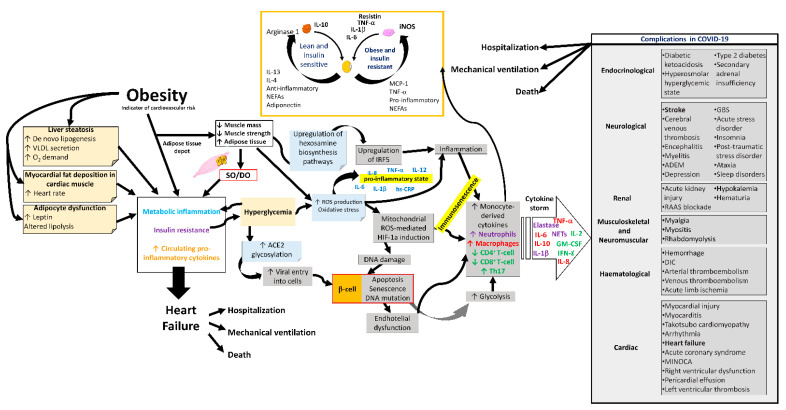
Complications in COVID-19 due to sarcopenic obesity and dynapenic obesity. Alterations in the metabolism, respiratory system, cardiovascular system, and immune system of the SO and/or DO patient with a propensity for complications during COVID-19. SO, sarcopenic obesity; DO, dynapenic obesity; ROS, reactive oxygen species; hs-CRP, high-sensitivity C-reactive protein; ACE2, angiotensin-converting enzyme 2; NETs, neutrophil extracellular traps; GM-CSF, granulocyte macrophage colony-stimulating factor; IFN-γ, gamma interferon; HIF-1a, hypoxia-inducible factor-1α; IRF5, interferon regulatory factor 5; VLDL, very low-density lipoprotein; ADEM, acute disseminated encephalomyelitis; GBS, Guillain–Barré syndrome; RAAS, renin–angiotensin–aldosterone system; DIC, disseminated intravascular coagulation; MINOCA, myocardial infarction with nonobstructive coronaries.

## Data Availability

Not applicable.

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
