# Peer review of "Associations of Dynapenic Obesity and Sarcopenic Obesity with the Risk of Complications in COVID-19"

_ijms, 2022, doi:10.3390/ijms23158277_

Round 1

Reviewer 1 Report

This review aims to evaluate the correlation among different types of obesity with the developing of severe Covid-19. Introduction is fine. Section 2 "obesity" is not significant. It could be as an introductory paragraph on section 3 (which should be section 2). 

Figure 2  must be improved in terms of image quality. 

Other than a few syntax errors, this is a concise review interesting  to readers. It does not add novel information but due to the pandemic and the constant "flow" of new information it should be published. 

Author Response

Point 1: Introduction is fine. Section 2 "obesity" is not significant. It could be as an introductory paragraph on section 3 (which should be section 2).

 Response 1: Section 2 is now in the introduction to section 3 and the number of sections has been reorganized.

Point 2: Figure 2 must be improved in terms of image quality.

Response 2: We have now improved and changed the figure.

Point 3: Other than a few syntax errors, this is a concise review interesting to readers. It does not add novel information but due to the pandemic and the constant "flow" of new information it should be published.

Response 3: Many thanks for your comments and suggestions for improving this manuscript, we have corrected syntax errors.

Reviewer 2 Report

The authors reviewed the mechanisms of SO/DO development and proposed that SO/DO is the risk of complications from Covid-19. Some comments below are for the authors to consider for revision of the article.

-        The statement on the study objective was unclear, lines 49-51, as this review did not ‘examine’ but proposed the mechanisms of comorbidity risk of Covid-19 patients with SO/DO.

-        Focusing more on associations of SO/DO with Covid-19 is essential. It might not be necessary to provide a section of review associations of SO/DO with CVD risk. Alternatively, considering complications and multisystemic diseases of Covid-19, those other than CVD may also need to be highlighted separately.

-        Lines 152-156, suggest that the age and sex about the significant findings in the cited study [66] should be indicated.

-        Lines 196-199, and 234-237, the statements were not written clearly.

-        Suggest consistent words and abbreviations throughout the article, e.g., lines 149, 150, Sarcopenic obesity, Dynapenic obesity, dynapenic-obesity, sarcopenic-obesity; line 212, dynapenic/abdominal obese (D/OA).

-        Please check the grammar and word spellings, e.g., lines 21, 27, 39/74/77 (BMI unit), and 81.

-        The full name of PPAR gamma in Figure 2 should be provided.

Author Response

Point 1: The statement on the study objective was unclear, lines 49-51, as this review did not ‘examine’ but proposed the mechanisms of comorbidity risk of Covid-19 patients with SO/DO.

 Response 1: We modified the objective of the study, lines 49-51.

Point 2: Focusing more on associations of SO/DO with Covid-19 is essential. It might not be necessary to provide a section of review associations of SO/DO with CVD risk. Alternatively, considering complications and multisystemic diseases of Covid-19, those other than CVD may also need to be highlighted separately.

Response 2: We have written more associations of SO/DO with COVID-19 and improvide the figure 3 related to complications in COVID-19 due to SO and DO.

Point 3: Lines 152-156, suggest that the age and sex about the significant findings in the cited study [66] should be indicated.

Response 3: We have indicated the age and sex about the significant findings in the cited study [66], now reference [65], lines 152-155

Point 4: Lines 196-199, and 234-237, the statements were not written clearly.

Response 4: We have rewritten the statements in lines 196-199 by lines 194-197, and 234-237 by lines 265-270.

Point 5: Suggest consistent words and abbreviations throughout the article, e.g., lines 149, 150, Sarcopenic obesity, Dynapenic obesity, dynapenic-obesity, sarcopenic-obesity; line 212, dynapenic/abdominal obese (D/OA).

Response 5: We have place consistent words and abbreviations throughout the article, considering corrections in Sarcopenic obesity, Dynapenic obesity, dynapenic-obesity, sarcopenic-obesity and dynapenic/abdominal obese on lines 37, 38, 48, 62, 63, 79, 82, 121, 124, 144, 148, 149, 152, 180, 183, 194, 206, 262, among other.

Point 6: Please check the grammar and word spellings, e.g., lines 21, 27, 39/74/77 (BMI unit), and 81.

Response 6: We have checked the grammar and word spellings throughout the article.

Point 7: The full name of PPAR gamma in Figure 2 should be provided.

Response 7: We have provided the full name of PPAR gamma in Figure 2. Many thanks for your comments and suggestions for improving this manuscript.

Round 2

Reviewer 2 Report

Thank you for your effort to revise the paper with considering the comments, particularly for adding paragraphs and Figure 3.

It would be good to go through the newly added paragraphs for grammar checking again. And the statement 'The association...14-54% [95].', lines 257-8 may not be correct.

Author Response

Point 1: Thank you for your effort to revise the paper with considering the comments, particularly for adding paragraphs and Figure 3

 Response 1: Many thanks for your comments and suggestions for improving this manuscript.

Point 2: It would be good to go through the newly added paragraphs for grammar checking again.

Response 2: We have check grammar again.

Point 3: The statement 'The association...14-54% [95].', lines 257-8 may not be correct.

Response 3:  We have rewritten the statement 'The association...14-54% [95].', lines 257-8 as 'It has been seen that in patients who have suffered from a stroke, a muscle mass and muscle strength reduction can occur, in addition to a fat deposition, which induces sarcopenia at a rate of 14 to 54% [95].', lines 258-61.
